# Density of *Aedes aegypti* and dengue virus transmission risk in two municipalities of Northwestern Antioquia, Colombia

**Wilber Gómez-Vargas** [1]*, **Paola Astrid Ríos-Tapias**[2], **Katerine Marin-Velásquez**[3], **Erika Giraldo-Gallo**[1], **Angela Segura-Cardona**[1], **Margarita Arboleda**[2]

1 Epidemiology and Biostatistics Group, Graduate School, Universidad CES, Medellín, Colombia, 2 Tropical Medicine Group, Colombian Institute of Tropical Medicine - Universidad CES, Apartadó, Colombia, 3 Tropical Medicine Group, Colombian Institute of Tropical Medicine - Universidad CES, Sabaneta, Colombia

* gomez.wilber@uces.edu.co

## Abstract

The high infestation of *Aedes aegypti* populations in Urabá, Antioquia, Colombia represents a risk factor for increased dengue morbidity and mortality. This study aimed to determine the risk of dengue transmission by estimating the population of *Ae. aegypti* using entomological indices, density of females per dwelling and inhabitant, and virological surveillance in two municipalities in Colombia. A cross-sectional study was conducted with quarterly entomological surveys in three neighborhoods of Apartadó and Turbo between 2021 and 2022. Aedes indices and vector density per dwelling and per inhabitant were calculated. The Kernel method was used for spatial analysis, and correlations between climatic variables and mosquito density were examined. Virus detection and serotyping in mosquitoes was performed using single-step reverse transcription polymerase chain reaction. The housing, reservoir, and Breteau indices were 48.9%, 29.5%, and 70.2%, respectively. The mean density of *Ae. aegypti* was 1.47 females / dwelling and 0.51 females / inhabitant. The overall visual analysis showed several critical points in the neighborhoods studied. There was significant correlation of vector density and relative humidity and precipitation in the neighborhoods 29 de noviembre and 24 de diciembre. Additionally, serotypes DENV-1 and DENV-2 were found. The overall indices for dwellings, reservoirs, and Breteau were lower than those recorded in 2014 in Urabá. The vector density results in this study were similar to those reported in other studies conducted in Latin America, and vector infection was detected. The Aedes and density indices are complementary, emphasizing the importance of continuous surveillance of *Ae. aegypti* to inform appropriate control strategies and prevent future dengue outbreaks in these municipalities.

## Introduction

The incidence of dengue and other arboviruses, such as chikungunya and Zika, in Colombia is closely associated with the density of the *Aedes aegypti* vector and its breeding sites, which can be found in urban, peri-urban, and rural areas where stagnant water is present [1–3].

**Funding:** This work was financially supported by the Ministry of Science, Technology, and Innovation of Colombia under project code 67882. The funder had no role in study design, data collection and analysis, decision to publish, or preparation of the manuscript.

**Competing interests:** The authors have declared that no competing interests exist.

Consequently, vector control measures remain the primary strategy for preventing dengue outbreaks until specific treatments and effective vaccines become available [3]. While *Aedes albopictus*, another mosquito species, has been found to carry dengue viruses in Colombia, it has not been implicated as a vector thus far [4–6] and there is no record of its presence in the Urabá region.

The Urabá region is particularly vulnerable to *Ae. aegypti* infestation due to its climate and high population density. Unregulated urbanization, coupled with inadequate sanitary conditions and deteriorating infrastructure, creates favorable conditions for the reproduction of *Ae. aegypti*. Although the subregion has achieved approximately 80% coverage of water and sewerage services, rural areas have significantly lower coverage, with less than 33% [7]. As a result, irregular service interruptions occur, leading to water scarcity during certain times of the year. Inhabitants resort to storing water in various containers, thereby facilitating the breeding of the vector.

Entomological surveillance plays a crucial role in monitoring and controlling *Ae. aegypti* populations. It involves systematic searches for breeding sites containing immature stages (larvae and pupae) as well as adult mosquitoes in urban and peri-urban areas. The objective is to gather information on infestation levels in each locality, serving as a fundamental component of vector control programs [8]. By conducting this surveillance, it becomes possible to assess the risk of arbovirus outbreaks, including dengue, chikungunya, and Zika, in different areas. This information is vital for directing appropriate and timely interventions for vector control [9].

The Colombian Ministry of Health and Social Protection has established the use of entomological or Aedes surveys as an official method for assessing the presence and abundance of *Ae. aegypti* [1]. These surveys use indices such as the dwelling index, reservoir index, and Breteau index. These indices are based on the number of houses visited and the presence or absence of immature mosquito stages. Despite operational challenges, such as difficulties in accessing residents' houses or locating hidden breeding sites like gutters or sewers, the results obtained through these indices are used to guide vector control efforts [10]. However, there are concerns regarding the reliability of these surveys as a method for evaluating *Ae. aegypti* populations. One limitation is that the indices obtained do not take into account the variation in adult mosquito production among different types of containers. Additionally, when vector infestation levels are low, these surveys may not accurately detect the presence of *Ae. aegypti* [11].

Among the various methods used in entomological surveillance, the collection of adult mosquitoes is particularly valuable as it provides information on the actual number of adult mosquitoes present. This data can be used to calculate the density of *Ae. aegypti* females per unit area or per inhabitant, which is a more accurate predictor of dengue occurrence [12, 13]. By focusing on adult mosquitoes, this method directly assesses the vector population responsible for disease transmission.

In addition, the use of geographic information systems (GIS) and spatial analysis has become increasingly important in predicting vector dispersion and implementing effective control measures. One useful technique in this regard is kernel density estimation (KDE), which involves smoothing and interpolating the distribution of points across the entire study area. KDE allows for the analysis of focal patterns and provides results that are easily interpretable [14]. By incorporating GIS and spatial analysis, researchers and public health authorities can gain valuable insights into the spatial distribution of *Ae. aegypti* populations and make informed decisions regarding vector control strategies.

The Urabá region has been experiencing a predominance of acute febrile syndrome, and dengue infection has been identified as a significant contributor to this syndrome since 2007 [15, 16]. A previous study conducted in the municipalities of Apartadó, Necoclí, and Turbo

revealed a high frequency of dengue infection, accounting for 37.3% of non-malarial febrile syndromes among the population [15].

Given the ongoing concerns regarding dengue transmission, the present study aimed to assess several indicators related to dengue risk. This included determining the Aedes indices, measuring the density of *Ae. aegypti* mosquitoes, analyzing their spatial distribution, and investigating the correlation between climatic variables (such as temperature, relative humidity, and precipitation) and vector density. Additionally, the study aimed to determine the rate of dengue virus infection. By examining these indicators in three specific neighborhoods of the Apartadó and Turbo municipalities in Antioquia, Colombia, the study aimed to establish the risk of dengue virus transmission in the region.

## Materials and methods

### Study area

This entomological study was conducted in the subregion of Urabá, specifically in three neighborhoods located in the municipalities of Apartadó and Turbo, which are part of the department of Antioquia in Colombia. The study was carried out between February 2021 and November 2022, in an area known for endemic dengue transmission.

Apartadó Municipality: The municipality of Apartadó (7º 53' 00" N; 76º 38' 00" W) covers an area of 600 km$^2$ and has a population of 112,396 inhabitants. The population density in Apartadó is 187 inhabitants per km$^2$, which is higher than the population densities of both the Urabá region (40 inhabitants per km$^2$) and the Antioquia department (93 inhabitants per km$^2$) [17]. The average temperature in Apartadó is 28˚C, and the municipality is situated at an altitude of 25 meters above sea level. Among the neighborhoods in Apartadó, the Serranía neighborhood (7˚ 53'7.65" N, 76˚ 38'2.51" W) was selected for this study due to high number of dengue cases in recent years (2018 to 2020: 82 cases) [18]. Serranía is characterized as a low to middle-income area with planned urbanization, high population density and basic infrastructure services such as piped water supply, sewerage, and regular garbage collection (Fig 1). This figure was constructed using Arc-Gis software and the area of the neighborhoods was delimited using the maps of Apartadó and Turbo, Antioquia, Colombia [19].

Nueva Colonia is a district located within the municipality of Turbo (7˚56'06.9 "N- 76˚ 43'01.6 "W) and is situated at an altitude of 2 meters above sea level. It functions as a population center with approximately 19,364 inhabitants. The population density in Nueva Colonia is 72 inhabitants per km$^2$, with 66.6% residing in 12 neighborhoods distributed among 3,377 houses, and the remaining 33.4% residing in 20 villages [20]. For this study, the neighborhoods of 29 de noviembre (7˚ 53'6.95" N, 76˚ 38'1.53" W) and 24 de diciembre (7˚ 56'1.93" N, 76˚ 42'9.87" W) were selected in Nueva Colonia. These neighborhoods have experienced a significant number of dengue cases and child deaths related to dengue [21] (Fig 1). They are characterized by unplanned urbanization, high population density, and low-income levels. The streets in these neighborhoods are unpaved, and there are puddles due to the lack of sewage systems. Additionally, the population lacks sufficient access to drinking water and regular solid waste collection services.

### Information on probable cases of dengue fever

The information on probable cases of dengue fever was obtained from the National Public Health Surveillance System (SIVIGILA) in Apartadó and Turbo for the period 2021–2022. Laboratory confirmation of dengue virus infection was done through RT-PCTR tests in the case of patients recruited ≤ five days of symptom onset and by detection of IgM antibodies by

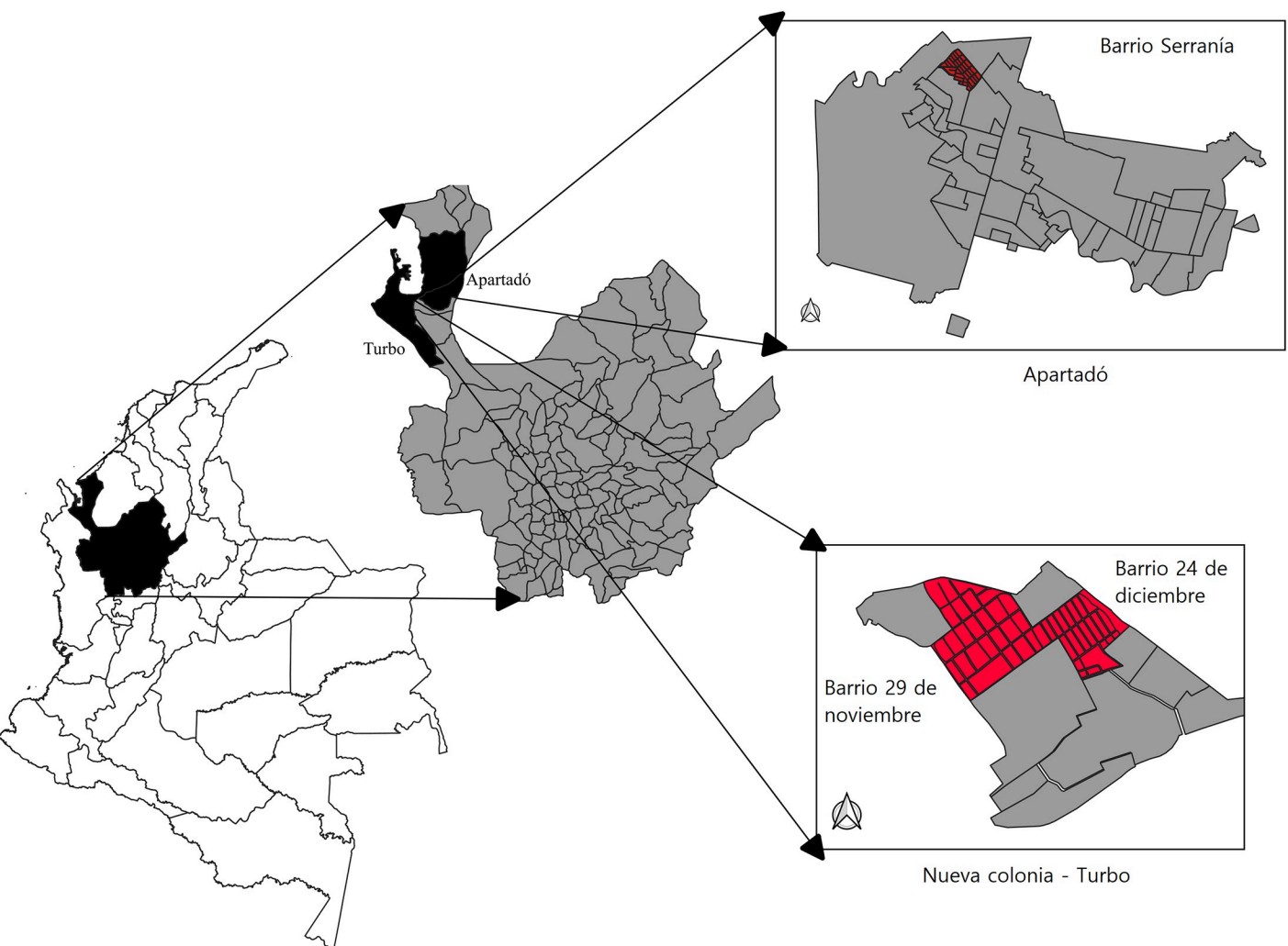

**Fig 1. Map of the study neighborhoods.** Serranía neighborhood, Apartadó, and 29 de noviembre and 24 de diciembre neighborhoods in the Nueva Colonia district, Turbo.

ELISA in those recruited from day six of symptom onset, in accordance with WHO criteria [22].

**Weather information.** Monthly records of precipitation (mm), relative humidity (%), and temperature (˚C) for the period 2021–2022 were obtained from two meterological stations of the Institute de Hidrología, Meteorología y Estudios Ambientales (IDEAM) located in Apartadó, Antioquia, Colombia. These data were used to perform correlation analyses between climatological variables and vector density in the three neighborhoods of the two municipalities. It should be noted that the weather information from the meteorological stations in the municipality of Turbo was not available. However, it is important to mention that the township of Nueva Colonia, Turbo, is geographically closer to the municipality of Apartadó than to the municipality of Turbo.

**Sample.** With a margin of error of 5%, a population proportion (P) of 50%, and a design effect of 1.1, a sample size of 1,608 dwellings was calculated. Considering a potential loss of 10%, a total of 1,518 dwellings were included in the entomological sampling across the three

neighborhoods. Out of these, 90 dwellings (5.6%) did not provide a response, resulting in a final sample size of 1,428 dwellings.

The sample size was calculated according to the following formula:

$$n = \frac{Z^2 N P (1 - P)}{N - 1 e^2 + Z^2 P (1 - P)}$$

Where;
Z = 1,96 (95%)
N = number of houses in the neighborhood
pq = variance
e = sampling error

The inspection of dwellings and the collection of adult mosquitoes were conducted eight times over a two-year period, from February 2021 to November 2022. A sampling interval of every 10th dwelling was used, and randomly generated numbers assigned to each dwelling were used to determine the starting point from which they were visited, following a clockwise direction. The households served as sampling units for the study.

## Entomological surveys

The inspection of houses was conducted using a Google® form, which automatically recorded the relevant information in a Microsoft Excel® workbook. The recorded data included the date, house number, owner's identification and name, number of inhabitants per house, types of breeding sites found, the positivity for immatures states (larvae and pupae), and the number of mosquitoes collected. The geographical coordinates of each inspected house were obtained using a GPS (Geographical Positioning System). The entomological surveys followed the operational guide provided by the World Health Organization for assessing the productivity of *Ae. aegypti* breeding sites [23]. The survey team consisted of a community leader, two research assistants, and a biologist. The team thoroughly inspected the dwellings, examining all potential water containers for both immature and adult mosquito stages. The intradomiciliary space, which included areas inside the dwelling such as bedrooms, living rooms, bathrooms, and kitchens, was inspected. Additionally, the peridomiciliary space, which encompassed the open areas of the property such as patios, gardens, and terraces, was also examined. A container containing larvae or pupae was considered positive for the presence of immature mosquito stages.

**Pupae collection.** The pupae found in the breeding sites were collected by using a standard plastic scoop and transferring them to a small container. From there, they were transferred to labeled plastic jars filled with water using a Pasteur pipette. In the laboratory, the collected pupae were moved to covered pupae emergence containers, where they were left until adult mosquitoes emerged.

For species identification adult mosquitoes were sacrificed by placing them in a -4 ˚C environment for 30 minutes. Morphological identification of the mosquitoes was used the key of Rueda (2004) [24] and was carried out using a stereomicroscope at the Entomology laboratory of the Instituto Colombiano de Medicina Tropical—CES University (ICMT-CES) in Apartadó, Antioquia, Colombia.

The emerged female *Ae. aegypti* mosquitoes were stored in pools containing 8 to 10 mosquitoes each. The mosquitoes were placed in 1.5 mL cryovial tubes with RNAlater and labeled according to the collection site. The samples were then preserved at 4 ˚C in the laboratory and later frozen at -80 ˚C for further processing [25].

## Collection of adult mosquitoes

The collection of adult mosquitoes was conducted in the same houses that were inspected. Mosquitoes were captured using Prokopack® aspirators (John W. Hock Co., Gainesville, FL, USA) for a duration of 20 minutes. The collection was carried out between 8:00 AM and 4:00 PM, following the sampling procedure described by Prokopec et al. [26]. After collection, captured mosquitoes were transferred to the Entomology laboratory of the ICMT-CES in Apartadó, Antioquia, Colombia. In the laboratory, the mosquitoes were identified and processed in the same manner as the emerged mosquitoes described previously.

## Mosquito processing and DENV detection

The pools of *Ae. aegypti* females were transported to the ICMT-CES laboratory in Medellín, Antioquia, Colombia for further processing. RNA extraction from each pool of mosquitoes was performed using the Quick-RNA™ Tissue/Insect Microprep kit from Zymo Research, following the manufacturer's protocol.

For the detection of dengue virus (DENV), the primers and probe suggested in the protocol by Gurukumar et al. [27] were used. To identify the serotype of the virus, the set of primers and primers described by Santiago et al. [28]. were used. All reverse transcription-polymerase chain reaction (RT-PCR) reactions were carried out using the SuperScript® III Platinum® one-step quantitative system. The reactions were performed on a Rotor-Gene Q thermocycler from QIAGEN®. The thermocycling parameters consisted of reverse transcription at 50 ˚C for 30 minutes, followed by denaturation at 95 ˚C for 2 minutes. Amplification and fluorescence detection were performed at 95 ˚C for 15 seconds and 60 ˚C for 1 minute, respectively, for a total of 40 cycles. Each probe's amplification curves were evaluated, and a value of $Cq \geq 37$ was considered negative for DENV presence.

## Data analysis

The data analysis was conducted using Microsoft Excel 2016 (Microsoft Corporation, Washington, USA). The first step involved calculating the proportion of mosquitoes by species based on the collected data. This information was used to determine the prevalence of *Ae. aegypti* and other mosquito species. Additionally, the data on dwellings with water containers were organized and presented in tables. To evaluate the risk of dengue virus (DENV) transmission, various dengue risk indicators were utilized. This included calculating the Aedes indices, which provide information on the density of *Ae. aegypti* mosquitoes in relation to the number of houses inspected. The mosquito density, spatial distribution, and their correlation with climatological variables such as precipitation, temperature, and relative humidity were also examined. Furthermore, the rate of DENV infection was determined to assess the likelihood of viral transmission.

The level of infestation by *Ae. aegypti* mosquitoes and the risk of dengue transmission were assessed using: (i) the housing index (HI), as the percentage of houses infested with larvae and/ or pupae; (ii) the deposit index (DI), as the percentage of water reservoirs infested with larvae and / or pupae; and (iii) the Breteau index (BI), as the number of positive reservoirs per 100 houses inspected in a specific location.

Based on WHO criteria, if the BI exceeds 50, HI exceeds 35, and DI exceeds 20, the risk of dengue transmission is considered high. When the BI falls between 5 and 50, it indicates a high density of *Ae. aegypti* mosquitoes and an increased risk of dengue transmission. Conversely, when the BI is below 5, HI is below 4, and DI is below 3, the likelihood of dengue transmission is considered low [29]. Additionally, indices based on female *Ae. aegypti* mosquitoes were used to calculate mosquito density for each dwelling.

To assess the positivity of *Ae. aegypti* mosquitoes, the percentage of infested houses with at least one female *Ae. aegypti* mosquito was calculated. The index of female *Ae. aegypti* mosquitoes per dwelling surveyed (Female/House) and per inhabitant per dwelling surveyed (Female/Inhabitant) were calculated (See S1 Appendix). To represent the spatial distribution of *Ae. aegypti* presence and overall density indices in each neighborhood, Kernel density relationships were employed. This technique utilizes Kernel density estimation to generate heat maps that highlight areas with higher concentrations of *Ae. aegypti* mosquitoes, helping to identify hotspots of vector infestation. For the construction of these maps (Fig 6), the traditional Aedes indices and the vector density indices obtained by the project with ArcGIS 10.5 software were used (see S1 Appendix).

All collected female mosquitoes were retained for DENV detection. The Minimum Infection Rate (MIR) was then calculated as the number of positive clusters (clusters of mosquitoes with DENV infection) divided by the total number of mosquitoes processed, multiplied by 1000. The MIR provides an estimate of the proportion of mosquitoes infected with DENV [30].

All statistical analyses were conducted using Jamovi version 2.2 [31], with a significance threshold (α) of 0.05. The Friedman test was employed to determine if there were significant differences in the density of *Ae. aegypti* mosquitoes among the neighborhoods studied. This non-parametric test is used when comparing multiple related samples. To analyze the potential relationship between climatic variables and vector density per neighborhood, both Spearman's and Pearson's correlation coefficients were used. The choice between these two correlation coefficients depended on the distribution of the data [32].

### Ethical considerations

The research protocol was submitted to the Institutional Bioethics Committee of the ICMT-CES in Medellín, and it received approval under Act 66 on June 14, 2019. Informed consent was obtained from each participant before conducting the entomological survey. The researchers sought written consent to enter their homes and carry out the entomological survey.

## Results

### Mosquito breeding sites and *Aedes aegypti* presence

The results of the entomological survey showed that out of the 1,518 homes inspected, 91.9% (1,394 homes) had water reservoirs. Among the three neighborhoods, the highest proportion of dwellings with water reservoirs was found in the 24 de diciembre neighborhood, at 99.2%. The neighborhood of 29 de noviembre had a similarly high proportion, with 99.0%. The Serranía neighborhood had a slightly lower proportion, with 93.7%.

Out of the 2,307 water reservoirs inspected, 40.5% (935) were found to have larvae and/or pupae of *Ae. aegypti* mosquitoes. Among the positive reservoirs, pools or low concrete tanks had the highest positivity rate, accounting for 25.6% (591/935) of the total. The 29 de noviembre neighborhood showed a higher infestation rate compared to other neighborhoods, at 18.2% (421 reservoirs) (Table 1).

The overall index of *Ae. aegypti* presence in dwellings (HI) was 48.9% (95% CI: 39.2–58.6 ± 24.2). The highest HI was recorded in the 24 de diciembre neighborhood (68.9%, 95% CI: 61.9–75.8 ± 10.1). The overall deposit index (DI) was 29.5% (95% CI: 25.1–34.0 ± 11.1), with the 24 de diciembre neighborhood having a higher DI (36.4%, 95% CI: 30.2–42.7 ± 9.01) compared to the other two neighborhoods. The overall Breteau index was 70.2 (95% CI: 53.6–86.9 ± 41.6), with the highest value in the 29 de noviembre neighborhood (100.4, 95% CI: 77.0–124.0 ± 33.7) (Fig 2).

**Table 1. Type and number of water reservoirs infested with larvae or pupae of *Aedes aegypti*.**

| Deposit type | Serranía | | 29 de noviembre | | 24 de diciembre | | Total | |
|---|---|---|---|---|---|---|---|---|
| | Positive deposits n (%) | Deposits n | Positive deposits n (%) | Deposits n | Positive Deposits n (%) | Deposits n | Positive deposits Total N (%) | Total deposits N |
| Low concrete tanks | 103 (4.5) | 493 | 257 (11.1) | 470 | 231 (10.0) | 396 | 591 (25.6) | 1359 |
| Plastic tanks | 22 (0.9) | 90 | 67 (2.9) | 370 | 41 (1.7) | 216 | 130 (5.6) | 676 |
| Bins | 13 (0.6) | 88 | 41 (1.7) | 263 | 44 (1.9) | 298 | 98 (4.2) | 649 |
| Tires | 3 (0.1) | 5 | 20 (0.9) | 68 | 10 (0.4) | 17 | 33 (1.4) | 90 |
| Others* | 24 (1.0) | 53 | 36 (1.6) | 141 | 23 (1.0) | 54 | 83 (3.6) | 248 |
| Total | 165 (7.1) | 729 | 421 (18.2) | 928 | 349 (15.1) | 650 | 935 (40.5) | 2307 |

* Bottles, disposable cups, pet drinking fountains.

**Species composition and density of mosquitoes collected.** At the end of entomological sampling in the Serranía neighborhood, Apartadó, and in the neighborhoods 29 de noviembre and 24 de diciembre in Nueva Colonia, Turbo, during the years 2021 and 2022, a total of 20,050 mosquitoes were collected. Of these, 72.8% (14,618) were *Culex* spp., 27.1% (5,408) were *Ae. aegypti* (2,564 females and 2,844 males, with a female to male ratio of 0.9:1.1), and 0.12% (24 individuals) were *Anopheles albimanus*. The highest proportion of *Ae. aegypti* mosquitoes was collected in the 24 de diciembre neighborhood, accounting for 12.3% (n = 2,462) (Table 2).

Dwellings infested with *Ae. aegypti* adults accounted for 57.7% (876) of all dwellings, and of which 48.9% (742) were infested with females and 47.8% (726) with males. The highest number of *Ae. aegypti* was observed in the 24 de diciembre neighborhood, with 514 mosquitoes captured in February 2022 (Fig 3).

The mean density of *Ae. aegypti* in the three neighborhoods across the eight samplings was 1.74 mosquitoes per dwelling (95% CI: 1.38–2.11 ± 1.29), and more males than females were collected: 2.02 males per dwelling (95% CI: 1.52–2.51 ± 1.23) versus 1.47 females (95% CI: 0.94–2.0 ± 1.32), a significant difference (Friedman's test: Q = 5.26, df = 1, p = 0.02) (Fig 4).

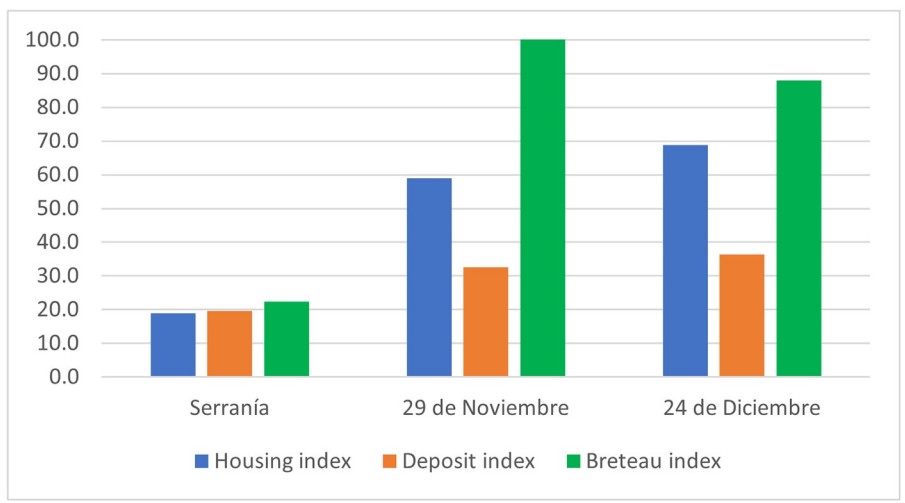

**Fig 2. Overall housing, deposits and Breteau indices by neighborhood.** 2021–2022.

**Table 2. Number and percentage of adult mosquitoes collected by neighborhood and species.**

| Neighborhood | Species | Female | Male | Total | % |
|---|---|---|---|---|---|
| | | (n) | (n) | (N) | |
| Serranía | *Aedes aegypti* | 555 | 745 | 1300 | 97.4 |
| | *Culex* spp. | 28 | 3 | 31 | 2.3 |
| | *Anopheles albimanus* | 4 | 0 | 4 | 0.3 |
| | Subtotal | 587 | 748 | 1335 | 6.7 |
| 29 de noviembre | *Aedes aegypti* | 816 | 830 | 1646 | 13.1 |
| | *Culex* spp. | 5040 | 5865 | 10905 | 86.8 |
| | *Anopheles albimanus* | 12 | 4 | 16 | 0.1 |
| | Subtotal | 5868 | 6699 | 12567 | 62.7 |
| 24 de diciembre | *Aedes aegypti* | 1193 | 1269 | 2462 | 12.3 |
| | *Culex* spp. | 1777 | 1905 | 3682 | 59.9 |
| | *Anopheles albimanus* | 3 | 1 | 4 | 0.1 |
| | Subtotal | 2973 | 3175 | 6148 | 30.7 |
| | Total | 9428 | 10622 | 20050 | |

Furthermore, there was no variation in the number of female mosquitoes captured per dwelling, as indicated by the coefficient of variation (CV) ≤ 1 (CV = SD/mean = 0.89).

The mean density of *Ae. aegypti* females and males per inhabitant in each of the dwellings in the three neighborhoods across the eight samplings was 0.53 (95% CI: 0.45–0.61 ± 0.28). More males than females were collected: 0.55 males / inhabitant (0.43–0.66 ± 0.28) compared to 0.51 females / inhabitant (95% CI: 0.39–0.63 ± 0.29). There were no significant differences between males and females in terms of density per inhabitant (Friedman's test: Q = 1.64, df = 1, p = 0.20) (Fig 5).

**Spatial distribution of *Aedes aegypti* density.** Fig 6 shows the spatial distribution of *Ae. aegypti* density in the surveyed dwellings during the entire study period (Eight cycles: 2021–2022) with Kernel density ratio maps. The maps highlight hot spots of the presence of immature stages and females of *Ae. aegypti* in the surveyed neighborhoods, indicating areas of high density.

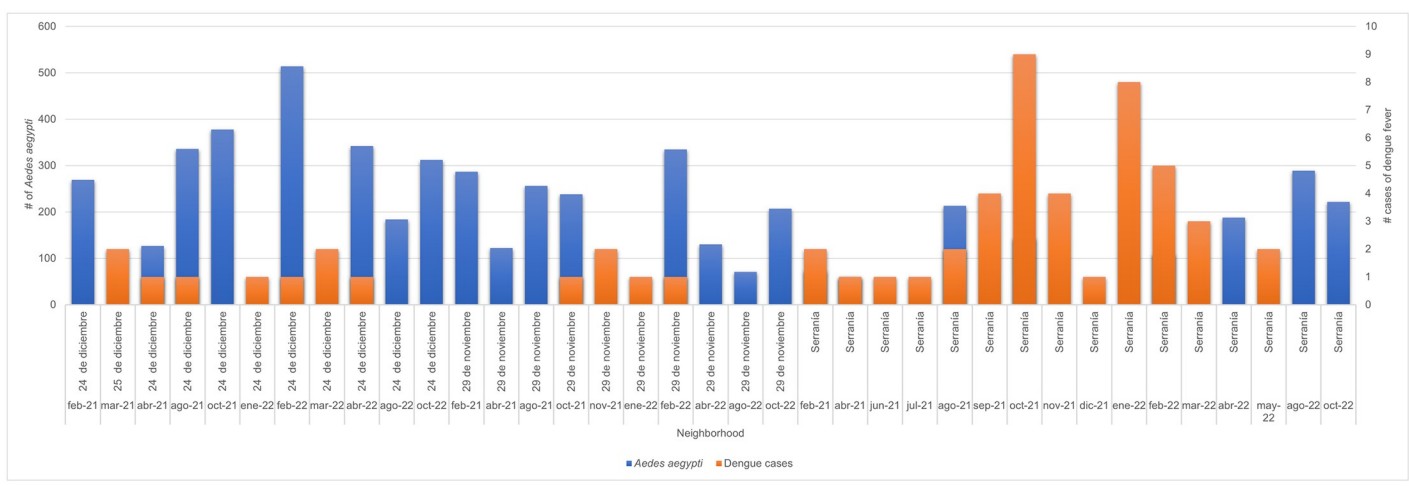

**Fig 3. Number of *Ae. aegypti* adults collected in the dwellings of the three study neighborhoods during the 2021–2022 period.**

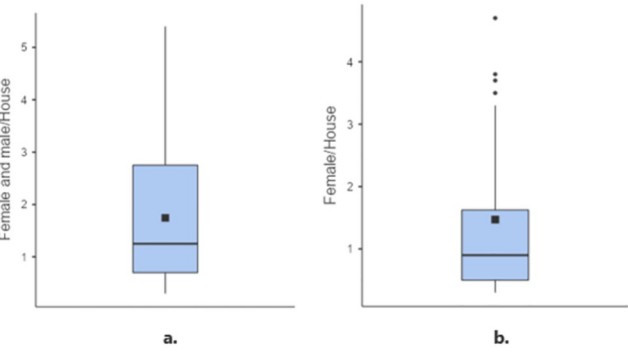

**Fig 4.** ***Aedes aegypti*** **adults collected per household in the three neighborhoods during the 2021–2022 period.** a. Number of *Aedes aegypti* adult females and males collected per household. b. Number of *Aedes aegypti* females collected per household. The box plot shows the median, interquartile range (IQR) and mean (■) values.

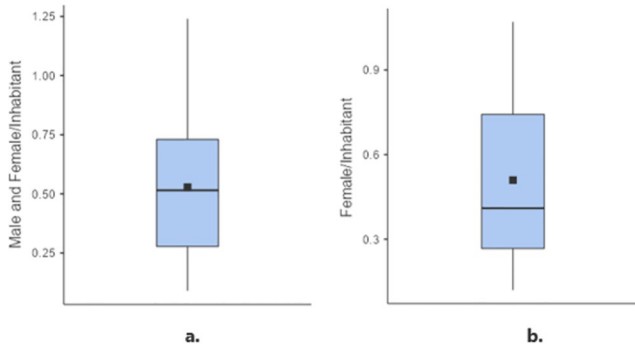

**Fig 5.** ***Aedes aegypti*** **adults collected per inhabitant in the three neighborhoods during the 2021–2022 period.** a. Number of *Aedes aegypti* adult females and males collected per inhabitant. b. Number of *Aedes aegypti* females collected per inhabitant. The box plot shows median, interquartile range (IQR) and mean (■) values.

**Relationship of climatic variables and vector density.** In the Serranía neighborhood, the vector density of *Ae. aegypti* females per inhabitant did not show a significant correlation with temperature (rs = 0.29, p = 0.48), relative humidity (rho = -0.200, p = 0.63), or precipitation (rs = -0.027, p = 0.95). However, in the 29 de noviembre and 24 de diciembre neighborhoods, there were significant correlations observed. In the 29 de noviembre neighborhood, there was a significant negative correlation between vector density and relative humidity (rs = -0.78, p = 0.022) and a significant positive correlation between vector density and precipitation (29 nov: rs = 0.750, p = 0.033; 24 dic: rs = 0.733, p = 0.038).

**Virus detection assays.** For the detection of Dengue virus (DENV) infection, a total of 249 groups or pools consisting of 2,131 *Ae. aegypti* females were tested using a one-step RT-PCR assay. Among these pools, three were found to be positive for DENV infection. The positive mosquitoes were collected from the 24 de diciembre neighborhood (2 pools) and the Serranía neighborhood (1 pool). Most of the infected females were found within the households. The minimum infection rate (MIR) for DENV was calculated as 1.4 (3 positive pools / 2,131 total individual mosquitoes processed × 1000), indicating the number of positive pools per 1,000 mosquitoes processed. The percentage of positive pools was calculated as 1.2% (3 pools / 249 pools).

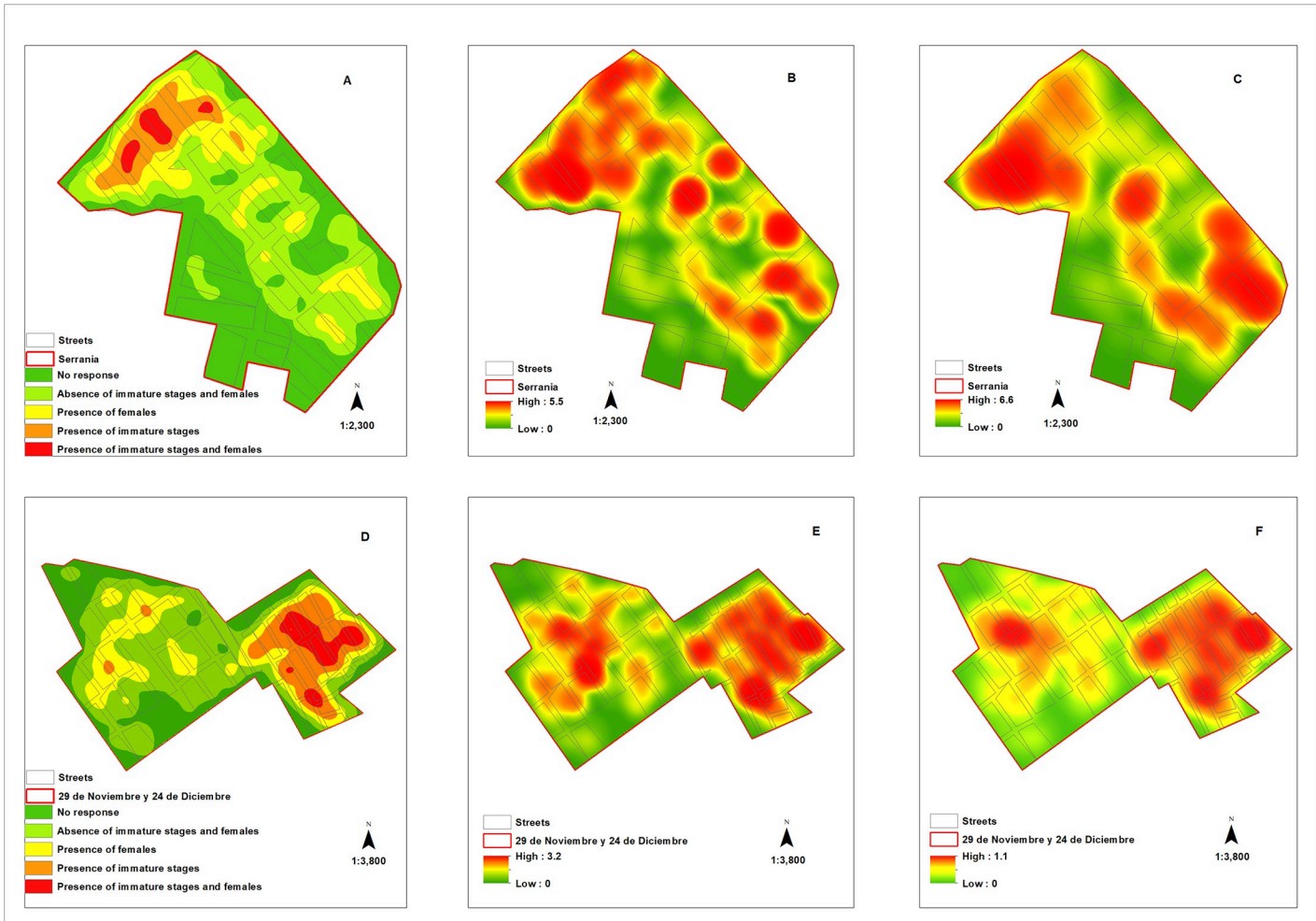

**Fig 6. Spatial distribution according to the presence of immature stages and females of *Ae. aegypti* in dwellings with their respective Kernel density ratio maps in general (eight cycles: 2021–2022).** Serranía (A: Presence, B: Females per dwelling, C: Females per inhabitant), 29 de noviembre and 24 de diciembre (D: Presence, E: Females per dwelling, F: Females per inhabitant).

During this period, 77 cases of dengue fever were recorded, 56 of which were reported by SIVIGILA and the total number of cases captured by the project was 33 and the number of positive dengue cases in humans was 21 in the neighborhoods mentioned, being 14, 5 and 2 cases in Serranía, 24 de diciembre and 29 de noviembre, respectively. Table 3 describes the

**Table 3. Description of dengue virus infection and distribution of serotypes in human patients and mosquitoes, according to neighborhood.**

| Neighborhood | Human cases notified SIVIGILA | Human cases captured Project | Human positive dengue cases | % human positivity | Patient serotypes | Number of pools analyzed | Number of Positive pools | % positive mosquitoes | Serotype |
|---|---|---|---|---|---|---|---|---|---|
| Serranía | 43 | 19 | 14 | 73.7 (14/19) | DENV-1 | 58 | 1 | 1.7 (1/58) | DENV-1 |
| 24-dic | 8 | 10 | 5 | 50.0 (5/10) | DENV-1 and DENV-2 | 102 | 2 | 2.0 (2/102) | DENV-1 and DENV-2 |
| 29-nov | 5 | 4 | 2 | 50.0 (2/4) | DENV-1 | 89 | 0 | 0.0 | Undetermined |
| Total | 56 | 33 | 21 | 63.6 (21/33) | DENV-1 and DENV-2 | 249 | 3 | 1.2 (3/249) | DENV-1 and DENV-2 |

number of human cases detected and the positivity ratio, as well as the serotypes found in patient samples and in mosquitoes.

## Discussion

The high Aedes indices observed in our study indicate a substantial presence of mosquito breeding sites within the study areas. The high-density rates of *Ae. aegypti* mosquitoes further highlights the potential for increased transmission of dengue fever in these neighborhoods. This observation aligns with previous studies conducted in the Antioquia region, which have reported similar patterns [33–36], for instance, a study conducted in Urabá documented a 32.5% [34] positivity rate in dwellings, which is lower than the 49% recorded in this study. It is worth noting that the comparison between our study and previous investigations should be interpreted with caution due to potential variations in study design, sampling methods, and geographical factors. However, the consistently high Aedes indices across different studies in the Antioquia region suggest a persistent and concerning issue of mosquito-borne disease transmission.

The high density of *Ae. aegypti* observed in this study is likely attributed to several factors, including the presence of numerous water reservoirs and the abundance of immature mosquitoes indoors. The shortage of water in the area may have prompted households to store water in tanks within their homes, leading to a higher number of breeding sites. Improper waste disposal and the prevalence of plastic tanks, buckets, and other water storage containers also contribute to the availability of suitable breeding sites for mosquitoes.

It is noteworthy that a significant proportion of water containers were found to harbor mosquito larvae and were left uncovered. This highlights the need for targeted mosquito control measures that prioritize breeding habitats, particularly focusing on water reservoirs suitable for long-term storage.

The overall housing, deposit, and Breteau indices observed in our study were lower than those reported by Alarcón et al. in 2014 in the Urabá region, specifically in one neighborhood of the municipality of Apartadó. Alarcón et al. documented values of 58.8%, 58.8%, and 111.8% for housing, deposits, and Breteau indices, respectively [33], These differences may be attributed to variations in sampling methods, climatic conditions, and local contexts between the two studies. Comparing the results to another study conducted in the El Manzanillo, Itagüí district in Antioquia, the housing and deposit indices were found to be similar, with values of 48.5% and 21.3%, respectively [36]. However, the Breteau index in that study was lower at 33%. These findings suggest some consistency in housing and deposit indices between the studies, while the variation in the Breteau index may be attributed to differences in mosquito breeding habitats and environmental factors.

This study highlights the elevated Aedes indices in the Nueva Colonia neighborhood, suggesting a higher risk of dengue virus transmission for its residents. The comparatively higher indices in the 24 de diciembre and 29 de noviembre neighborhoods, in contrast to the Serranía neighborhood, can be attributed to various factors. These neighborhoods, characterized by lower socioeconomic status, may have intra-domiciliary and peridomiciliary environments with inadequate water storage practices, such as low tanks, plastic containers, and uncovered garbage cans, which provide favorable breeding sites for mosquitoes.

These observations emphasize the importance of implementing targeted mosquito control measures in these neighborhoods to mitigate the risk of DENV transmission. Effective control practices may include regular inspection and elimination of potential breeding sites, promoting proper water storage practices, and raising awareness among residents about the importance of maintaining clean and mosquito-free environments.

The relationship between human density and vector density, particularly the number of *Ae. aegypti* females, is an important indicator of the risk of dengue transmission, as highlighted by previous studies [37]. This human-vector proximity has been reported by different authors. For instance, a study conducted in Taiwan in 1994 reported a density of 0.07 females/inhabitant [38], while studies in Colombia described densities of 0.5 females/inhabitant [39] and 2.1 females/ dwelling [32]. In other studies conducted in Ibagué, Colombia, Manta, Ecuador, and Posadas, Argentina, densities of 2.2, 1.13, and 1.80 females/dwelling, respectively, were observed, with an average of 1.71 females/dwelling across the three countries [40]. Similarly, a study conducted in Brazil reported densities of 1.60 females/dwelling and 0.42 females/inhabitant [37]. The present study found densities of 1.47 females/dwelling and 0.51 females/inhabitant, with the majority of female *Ae. aegypti* mosquitoes collected in the Nueva Colonia neighborhood, particularly in the 24 de diciembre neighborhood. These findings align with the results of previous studies conducted in Latin American countries, suggesting a comparable risk of dengue transmission in these areas.

The findings of this study indicate that the vector density, specifically the number of *Ae. aegypti* females per inhabitant, in the Serranía neighborhood is not significantly correlated with the environmental variables of temperature, humidity, or precipitation, which has a good water supply all year round. This suggests that factors other than these climatic variables may be influencing the vector density in this neighborhood, such as the presence of the vector outside the homes such as disposable containers, tires, among others. On the other hand, the neighborhoods of 29 de noviembre and 24 de diciembre show a significant correlation between vector density and relative humidity as well as precipitation. This implies that the abundance of *Ae. aegypti* females in these neighborhoods is influenced by the levels of humidity and rainfall. In these neighborhoods, due to the poor water supply, the inhabitants are forced to store water throughout the year, especially in dry weather. These environmental conditions may create suitable breeding habitats for mosquitoes, resulting in increased vector density.

Additionally, when considering the number of females per dwelling, only the neighborhood of 24 de diciembre shows a significant correlation with precipitation. This neighborhood, which also recorded the highest number of dengue cases among the two neighborhoods studied in the Nueva Colonia locality, may be more susceptible to dengue transmission due to the combination of high vector density and favorable environmental conditions. That is why the rains favor the increase of breeding sites and cases of dengue. In dry periods and low temperatures, vectors decrease, a situation that does not stop the transmission of the disease, due to the active biting behavior of the vector that favors the spread of the virus [41]. It can be observed that the impact of rainfall varied according to the neighborhood, this may be due to the differences in the types of water storage containers with *Ae. aegypti*'s larvae that affects the density of the adult mosquito, as was the case with a study conducted in Ecuador by Stewart-Ibarra et al. [42].

Being located near the equator, as suggested by Carrasquilla et al., [32], Colombia experiences relatively stable temperatures throughout the year, which may explain the lack of significant correlation between temperature and vector density in the Serranía neighborhood. Nevertheless, it is important to note that numerous studies have demonstrated the influence of climatic variables on dengue transmission. Factors such as temperature, humidity, and precipitation have been identified as key drivers of mosquito population dynamics and dengue transmission patterns in various regions [43].

In Brazil, when analyzing the correlation between the total number of insects collected and *Ae. aegypti* house indices (HI) in relation to climatic variables of temperature, humidity, rainfall, among others, regardless of locality, these variables had a direct influence on the

distribution and presence of mosquitoes [44, 45]. Therefore, it is important to highligth that mosquito populations can change, to better adapt to local environmental conditions [46, 47].

The identification of hotspot foci with a specific pattern in the spatial distribution of immature and adult female *Ae. aegypti* aligns with previous research highlighting the focal nature of *Ae. aegypti* distribution. It is widely observed that the distribution of *Ae. aegypti* mosquitoes tends to be highly localized, forming clusters or hotspots of high vector abundance within small clusters of houses.

These hotspots of high vector abundance may exhibit temporal instability, meaning they can vary in location and intensity over time [48, 49].

The findings of this study suggest that the use of indicators of female density per house and per inhabitant provides a more accurate representation of the spatial distribution of *Ae. aegypti* compared to using presence / absence data of immature stages and females. This is in contrast to a recent study conducted in Brazil, where no consistent pattern in the spatial distribution of *Ae. aegypti* eggs, larvae, or adults was observed using separate entomological indicators [50]. The example from Apartadó and Carepa, where hotspots of *Ae. aegypti* infestation were identified based on egg density, highlights the utility of using specific entomological indicators for targeted vector control activities [33]. Similarly, in Medellín, spatial analysis techniques such as Kernel density and hot spot analysis were employed to identify areas with a higher presence of the vector, aiding in the implementation of effective vector control strategies [51].

The laboratory analysis conducted in this study revealed the presence of DENV serotypes 1 and 2 in mosquitoes collected from the 24 de diciembre and Serranía neighborhoods. This finding is consistent with previous entomo-virological studies conducted in Colombia, where all four DENV serotypes have been detected, with serotypes 1 and 2 being the most predominant [6, 30, 32, 52]. Although the infection rates observed in this study were relatively low, with a minimum infection rate of 1.4 and a positivity rate of 1.2%, the high density and large number of infested water reservoirs indicate a significant risk of DENV transmission in the study areas. It is worth noting that no studies investigating vector infection in the Urabá region were found in the literature review conducted. The presence of DENV infection in mosquitoes can be influenced by various factors, including rainfall, the number of mosquitoes tested, and the circulation of DENV among the human population during the study period. Despite the low number of laboratory-confirmed dengue cases among residents of the study neighborhoods, the detection of DENV serotypes 1 and 2 in both human samples mosquito pools in the 24 de diciembre neighborhood, as well as the presence of DENV serotype 1 in both humans and adult mosquito pools in the 24 de diciembre and Serranía neighborhoods, is noteworthy. Of the 21 dengue cases identified, 12 positive cases were identified by RT-PCR, of which the DEN-1 serotype was the most frequent with 75% (9/12), followed by the DEN-2 serotype with 25% (3/12). The remaining nine were identified by detection of IgM antibodies using the Elisa technique and six of these also had the presence of NS1 antigen.

Recent studies conducted in Medellín, Antioquia, analyzed pools of *Aedes aegypti* mosquitoes and found a 13% positivity rate for DENV [53], however there is a lack of studies investigating DENV positivity in mosquitoes in Urabá.

These findings underscore the importance of entomo-virological surveillance as part of the routine surveillance of municipalities which is a useful tool to identify high-risk areas for virus transmission and an epidemiological alert system to directly control critical areas of activity for vector control especially because people circulating or living in places where these infected mosquitoes are identified are exposed to bites most of the day [54–57]. Strategies for virological surveillance of Aedes mosquitoes commonly focus on females because females are hematophagous, whereas males are not [58]. Although DENV is transmitted primarily by the bite of female mosquitoes infected with human blood (horizontal transmission), both vertical (in

which the infected mosquito is able to transmit the virus to its progeny) and sexual (when an infected male transmits the virus to the female during copulation) transmission have been suggested as an important mechanism for maintaining arbovirus circulation in vector populations [59–61]. However, the challenges for the incorporation of entomo-virological surveillance in dengue control programs must have the necessary infrastructure, trained personnel and political (financial) support to implement it and the importance of evaluating its impact on the disease burden in the short, medium, and long term. Its inclusion in control programs should be assessed according to local capacities and the integrated use of other control tools [62]. Detecting DENV infection in mosquitoes provides valuable information for understanding the transmission dynamics and implementing targeted vector control measures to mitigate the risk of dengue transmission.

## Conclusions

This study highlights several important findings regarding the risk of dengue transmission and the presence of *Ae. aegypti* mosquitoes in the neighborhoods under study. The high infestation rates of water reservoirs with mosquito larvae and pupae indicate a conducive environment for mosquito breeding. The presence and density of *Ae. aegypti* mosquitoes, particularly in the Nueva Colonia neighborhood, suggest an increased risk of DENV transmission in these areas.

The detection of females carrying DENV serotypes 1 and 2 in the study neighborhoods, along with the correlation with dengue cases among residents, further supports the likelihood of local transmission. This emphasizes the importance of targeted vector control measures to reduce the risk of dengue transmission in these communities.

The use of entomological indicators, specifically the presence and density of adult female mosquitoes, provides valuable complementary information to traditional Aedes indices. Integrating these indicators into vector surveillance efforts can enhance the understanding of dengue and other arbovirus transmission dynamics, enabling more effective monitoring and control strategies.

### Limitations of the study

The study was conducted in only three neighborhoods in two municipalities of Urabá, which may limit the generalizability of the findings to other areas within the region. These neighborhoods were selected based on their high dengue burden and specific urban and demographic characteristics, which may differ from other municipalities in Urabá.

The study's timeframe and duration may also be considered a limitation. The data collection period and the number of monitoring cycles may not capture the full seasonal dynamics and fluctuations in vector density and viral activity, potentially leading to an incomplete representation of the transmission patterns.

## Supporting information

**S1 Appendix.**
(XLSX)

## Acknowledgments

The authors would like to extend their heartfelt thanks to the residents who generously granted access to their homes for the purpose of entomological sampling. The authors also acknowledge the invaluable support and assistance provided by the Health Secretaries of Apartadó and Turbo in Antioquia, Colombia.

## Author Contributions

**Conceptualization:** Wilber Gómez-Vargas, Paola Astrid Ríos-Tapias, Katerine Marin-Velásquez.

**Data curation:** Wilber Gómez-Vargas, Paola Astrid Ríos-Tapias, Katerine Marin-Velásquez.

**Formal analysis:** Wilber Gómez-Vargas, Paola Astrid Ríos-Tapias, Katerine Marin-Velásquez, Angela Segura-Cardona, Margarita Arboleda.

**Funding acquisition:** Paola Astrid Ríos-Tapias, Margarita Arboleda.

**Investigation:** Wilber Gómez-Vargas, Paola Astrid Ríos-Tapias, Katerine Marin-Velásquez.

**Methodology:** Wilber Gómez-Vargas, Paola Astrid Ríos-Tapias, Katerine Marin-Velásquez.

**Project administration:** Paola Astrid Ríos-Tapias.

**Supervision:** Margarita Arboleda.

**Validation:** Erika Giraldo-Gallo, Angela Segura-Cardona.

**Writing – original draft:** Wilber Gómez-Vargas, Paola Astrid Ríos-Tapias, Katerine Marin-Velásquez.

**Writing – review & editing:** Wilber Gómez-Vargas, Paola Astrid Ríos-Tapias, Katerine Marin-Velásquez, Erika Giraldo-Gallo, Angela Segura-Cardona, Margarita Arboleda.

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
