## [Decision Letter · Decision Letter 0]

3 Sep 2023

PONE-D-23-22160Density of Aedes aegypti and Dengue Virus Transmission Risk in Two Municipalities of Northwestern Antioquia, ColombiaPLOS ONE

Dear Dr. Gómez-Vargas,

Thank you for submitting your manuscript to PLOS ONE. After careful consideration, we feel that it has merit but does not fully meet PLOS ONE’s publication criteria as it currently stands. Therefore, we invite you to submit a revised version of the manuscript that addresses the points raised during the review process.

We look forward to receiving your revised manuscript.

Kind regards,

Harapan Harapan, MD, PhD

Academic Editor

PLOS ONE

2. We note that Figures 1 and 6 in your submission contain [map/satellite] images which may be copyrighted. All PLOS content is published under the Creative Commons Attribution License (CC BY 4.0), which means that the manuscript, images, and Supporting Information files will be freely available online, and any third party is permitted to access, download, copy, distribute, and use these materials in any way, even commercially, with proper attribution. For these reasons, we cannot publish previously copyrighted maps or satellite images created using proprietary data, such as Google software (Google Maps, Street View, and Earth). For more information, see our copyright guidelines: http://journals.plos.org/plosone/s/licenses-and-copyright.

a. You may seek permission from the original copyright holder of Figures 1 and 6 to publish the content specifically under the CC BY 4.0 license. 

Reviewers' comments:

Reviewer's Responses to Questions

**Comments to the Author**

1. Is the manuscript technically sound, and do the data support the conclusions?

Reviewer #1: Yes

Reviewer #2: Yes

2. Has the statistical analysis been performed appropriately and rigorously? 

Reviewer #1: Yes

Reviewer #2: I Don't Know

3. Have the authors made all data underlying the findings in their manuscript fully available?

Reviewer #1: Yes

Reviewer #2: Yes

4. Is the manuscript presented in an intelligible fashion and written in standard English?

Reviewer #1: Yes

Reviewer #2: Yes

5. Review Comments to the Author

Reviewer #1: In this article, authors present entomological and virological data related to dengue virus and its transmission in three neighborhoods in Colombia. Though findings are largely descriptive, the information adds to the existing knowledge on vector indices in Latin America that affect dengue in the region.

The article is well-written and easy to understand.

I have included several comments in the attached manuscript for authors' consideration.

Reviewer #2: The author's report regarding dengue fever should include an explanation of the laboratory test that was used to determine confirmed dengue cases, either by a serological test (NS-1, IgG, IgM) or viral load. The author should also elaborate on which WHO criteria that was used. The author needs to be careful when citing references. For example, in line 178 references number 20 and 21 did not refer to the WHO diagnostic criteria.

Rather than describing the result, the authors should explain the different findings in the correlation of temperature, humidity, and precipitation with vector density, in Serranía, 29 de noviembre, and 24 de diciembre. Explain why in Serranía there was no correlation between vector density with temperature, humidity, and precipitation, while on 29 de noviembre, and 24 de diciembre significant correlation was observed.

The authors need to elaborate on the inconsistency between Table 3 and the paragraph preceding the Table. The total cases reported by SIVIGILA in the paragraph preceding the table and table were consistent, but there was inconsistency in the Total cases captured by the project in the paragraph preceding the table which was reported to be 21 while in Table 3 the total cases captured by the project was 33 and the human positive dengue cases was 21. The author also needs to put the laboratory test that was used to confirm dengue infection in these people.

The author should consider putting the sample size formula in The Sample subsection line 194 so that the study could easily reproduced by others who want to do similar research in different countries.

6. PLOS authors have the option to publish the peer review history of their article (what does this mean?). If published, this will include your full peer review and any attached files.

Reviewer #1: No

Reviewer #2: No

---

## [Author Response · Author response to Decision Letter 0]

18 Oct 2023

Medellín, Colombia, October 18, 2023

Harapan Harapan, MD, PhD

Academic Editor

PLOS ONE

Subject: Response to comments done to the manuscript: Density of Aedes aegypti and Dengue Virus Transmission Risk in Two Municipalities of Northwestern Antioquia, Colombia. 

Journal requirements

 PLOS ONE style requirements were met, including those related to file naming.

 Figures 1 and 6 do not contain [map/satellite] images that may be copyrighted.

In relation to figure 1, in the text on lines 151 to 154 it is explained that: “This figure was constructed using Arc-Gis software and the area of the neighborhoods was delimited using the maps of Apartadó and Turbo, Antioquia, Colombia”.

In relation to figure 6, in the text on lines 330 to 333 it is explained that: “For the construction of these maps (Figure 6), the traditional Aedes indices and the vector density indices obtained by the project with ArcGIS 10.5 software were used (see supplementary information S1).”.

Response to reviewer's comments 1

 Suggest to use Aedes instead of aedic throughout the manuscript: Suggested changes were made throughout the manuscript.

 80% - Please check throughout the text: Suggested changes were made throughout the manuscript.

 Please state the time or duration of this occurrence: 

“The Urabá region has been experiencing a predominance of acute febrile syndrome, and dengue infection has been identified as a significant contributor to this syndrome since 2007 (15,16).” (Lines in the text of 146 to 148).

 Any citations or data to support this statement? A quote and data were included that supports this statement. “…was selected for this study due to high number of dengue cases in recent years (2018 to 2020: 82 cases) (18).”. (Lines in the text of 146 to 148).

 Positivity for immatures?: the positivity for immatures states (larvae and pupae). (Lines in the text of 223 to 224).

 Please state which taxonomic keys were used for the identification: was used the key of Rueda (2004). (Line 250).

 Suggest to add the number of human cases reported for each period in the secondary axis of Fig 3 to demonstrate the correlation between the mosquito abundance and human cases in the study areas: The number of human cases reported for each period were added on the X axis in Fig 3.

 Is this DEN-1 and DEN-2?: The table was corrected.

 Please discuss these contrasting findings on climatic variable in a broader context based on what has previously been recorded in literature, particularly in the Americas.

“..which has a good water supply all year round”. (Lines in the text of 580 to 581).

 “…such as the presence of the vector outside the homes such as disposable containers, tires, among others”. (Lines in the text of 583 to 584).

“In these neighborhoods, due to the poor water supply, the inhabitants are forced to store water throughout the year, especially in dry weather”. (Lines in the text of 590 to 591).

“That is why the rains favor the increase of breeding sites and cases of dengue. In dry periods and low temperatures, vectors decrease, a situation that does not stop the transmission of the disease, due to the active biting behavior of the vector that favors the spread of the virus (41). It can be observed that the impact of rainfall varied according to the neighborhood, this may be due to the differences in the types of water storage containers with Ae. aegypti's larvae that affects the density of the adult mosquito, as was the case with a study conducted in Ecuador by Stewart-Ibarra et al. (42)”. (Lines in the text of 600 to 609).

“In Brazil, when analyzing the correlation between the total number of insects collected and Ae. aegypti house indices (HI) in relation to climatic variables of temperature, humidity, rainfall, among others, regardless of locality, these variables had a direct influence on the distribution and presence of mosquitoes (44,45). Therefore, it is important to highligth that mosquito populations can change, to better adapt to local environmental conditions (46,47)” (Lines in the text of 619 to 625).

 Please provide additional information on the proportion of DENV-1 and DENV-2 among human samples in the study area. This data will strengthen the vector findings: 

Additional information was as follows: “Of the 21 dengue cases identified for the project, 12 positive cases were identified by RT-PCR, of which the DEN-1 serotype was the most frequent with 75% (9/12), followed by the DEN-2 serotype with 25% (3/12). The remaining nine were identified by detection of IgM antibodies using the Elisa technique and six of these also had the presence of NS1 antigen.”. (Lines in the text of 669 to 674). 

 Please also discuss the challenges in conducting entomo-virological surveillance. Even though it is essential, many challenges affect the utility of this approach:

Lines in the text of 679 to 700:

“These findings underscore the importance of entomo-virological surveillance as part of the routine surveillance of municipalities which is a useful tool to identify high-risk areas for virus transmission and an epidemiological alert system to directly control critical areas of activity for vector control especially because people circulating or living in places where these infected mosquitoes are identified are exposed to bites most of the day (48–51). Strategies for virological surveillance of Aedes mosquitoes commonly focus on females because females are hematophagous, whereas males are not (52). Although DENV is transmitted primarily by the bite of female mosquitoes infected with human blood (horizontal transmission), both vertical (in which the infected mosquito is able to transmit the virus to its progeny) and sexual (when an infected male transmits the virus to the female during copulation) transmission have been suggested as an important mechanism for maintaining arbovirus circulation in vector populations (53–55). However, the challenges for the incorporation of entomo-virological surveillance in dengue control programs must have the necessary infrastructure, trained personnel and political (financial) support to implement it and the importance of evaluating its impact on the disease burden in the short, medium, and long term. Its inclusion in control programs should be assessed according to local capacities and the integrated use of other control tools (56)”. 

Responses to reviewer's comments 2

 The author's report regarding dengue fever should include an explanation of the laboratory test that was used to determine confirmed dengue cases, either by a serological test (NS-1, IgG, IgM) or viral load. The author should also elaborate on which WHO criteria that was used. The author needs to be careful when citing references. For example, in line 178 references number 20 and 21 did not refer to the WHO diagnostic criteria.

“Laboratory confirmation of dengue virus infection was done through RT-PCTR tests in the case of patients recruited ≤ five days of symptom onset and by detection of IgM antibodies by ELISA in those recruited from day six of symptom onset, in accordance with WHO criteria (22).” The reference was updated. (Lines in the text of 177 to 181). 

 Rather than describing the result, the authors should explain the different findings in the correlation of temperature, humidity, and precipitation with vector density, in Serranía, 29 de noviembre, and 24 de diciembre. Explain why in Serranía there was no correlation between vector density with temperature, humidity, and precipitation, while on 29 de noviembre, and 24 de diciembre significant correlation was observed.

 “..which has a good water supply all year round”. (Lines in the text of 580 to 581).

 “…such as the presence of the vector outside the homes such as disposable containers, tires, among others”. (Lines in the text of 583 to 584).

“In these neighborhoods, due to the poor water supply, the inhabitants are forced to store water throughout the year, especially in dry weather”. (Lines in the text of 590 to 591).

“That is why the rains favor the increase of breeding sites and cases of dengue. In dry periods and low temperatures, vectors decrease, a situation that does not stop the transmission of the disease, due to the active biting behavior of the vector that favors the spread of the virus (41). It can be observed that the impact of rainfall varied according to the neighborhood, this may be due to the differences in the types of water storage containers with Ae. aegypti's larvae that affects the density of the adult mosquito, as was the case with a study conducted in Ecuador by Stewart-Ibarra et al. (42)”. (Lines in the text of 600 to 609).

“In Brazil, when analyzing the correlation between the total number of insects collected and Ae. aegypti house indices (HI) in relation to climatic variables of temperature, humidity, rainfall, among others, regardless of locality, these variables had a direct influence on the distribution and presence of mosquitoes (44,45). Therefore, it is important to highligth that mosquito populations can change, to better adapt to local environmental conditions (46,47)” (Lines in the text of 619 to 625).

 The authors need to elaborate on the inconsistency between Table 3 and the paragraph preceding the Table. The total cases reported by SIVIGILA in the paragraph preceding the table and table were consistent, but there was inconsistency in the Total cases captured by the project in the paragraph preceding the table which was reported to be 21 while in Table 3 the total cases captured by the project was 33 and the human positive dengue cases was 21. The author also needs to put the laboratory test that was used to confirm dengue infection in these people.

 The paragraph was corrected:

“During this period, 77 cases of dengue fever were recorded, 56 of which were reported by SIVIGILA and the total number of cases captured by the project was 33 and the number of positive dengue cases in humans was 21 in the neighborhoods mentioned, being 14, 5 and 2 cases in Serranía, 24 de diciembre and 29 de noviembre, respectively.” (Lines in the text of 485 to 490). 

 The author should consider putting the sample size formula in The Sample subsection line 194 so that the study could easily reproduced by others who want to do similar research in different countries.

The sample size was calculated according to the following formula: 

=(Z^2 NP(1-P))/(N-1e^(2 )+Z^2 P(1-P))

Where;

Z =1,96 (95%)

N= number of houses in the neighborhood

pq= variance 

e= sampling error

(Lines in the text of 203 to 209). 

 Kind regards;

Wilber Gómez-Vargas

Gomez.wilber@uces.edu.co

---

## [Decision Letter · Decision Letter 1]

20 Nov 2023

Density of Aedes aegypti and Dengue Virus Transmission Risk in Two Municipalities of Northwestern Antioquia, Colombia

PONE-D-23-22160R1

Dear Dr. Gómez-Vargas,

We’re pleased to inform you that your manuscript has been judged scientifically suitable for publication and will be formally accepted for publication once it meets all outstanding technical requirements.

Kind regards,

Harapan Harapan, MD, PhD

Academic Editor

PLOS ONE

Additional Editor Comments (optional):

Reviewers' comments:

Reviewer's Responses to Questions

**Comments to the Author**

1. If the authors have adequately addressed your comments raised in a previous round of review and you feel that this manuscript is now acceptable for publication, you may indicate that here to bypass the “Comments to the Author” section, enter your conflict of interest statement in the “Confidential to Editor” section, and submit your "Accept" recommendation.

Reviewer #1: All comments have been addressed

Reviewer #2: All comments have been addressed

2. Is the manuscript technically sound, and do the data support the conclusions?

Reviewer #1: Yes

Reviewer #2: Yes

3. Has the statistical analysis been performed appropriately and rigorously? 

Reviewer #1: Yes

Reviewer #2: Yes

4. Have the authors made all data underlying the findings in their manuscript fully available?

Reviewer #1: Yes

Reviewer #2: Yes

5. Is the manuscript presented in an intelligible fashion and written in standard English?

Reviewer #1: Yes

Reviewer #2: Yes

6. Review Comments to the Author

Reviewer #1: Authors have addressed the comments satisfactorily. Pls make sure to go through the manuscript to correct minor grammatical and syntax errors.

Reviewer #2: (No Response)

7. PLOS authors have the option to publish the peer review history of their article (what does this mean?). If published, this will include your full peer review and any attached files.

Reviewer #1: No

Reviewer #2: No

---

## [Editor Report · Acceptance letter]

3 Jan 2024

PONE-D-23-22160R1 

PLOS ONE

Dear Dr. Gómez-Vargas, 

I'm pleased to inform you that your manuscript has been deemed suitable for publication in PLOS ONE. Congratulations! Your manuscript is now being handed over to our production team.

Kind regards, 

on behalf of

Dr. Harapan Harapan 

Academic Editor

PLOS ONE